# Antibiotic Therapy Duration for Multidrug-Resistant Gram-Negative Bacterial Infections: An Evidence-Based Review

**DOI:** 10.3390/ijms26146905

**Published:** 2025-07-18

**Authors:** Andrea Marino, Egle Augello, Carlo Maria Bellanca, Federica Cosentino, Stefano Stracquadanio, Luigi La Via, Antonino Maniaci, Serena Spampinato, Paola Fadda, Giuseppina Cantarella, Renato Bernardini, Bruno Cacopardo, Giuseppe Nunnari

**Affiliations:** 1Unit of Infectious Diseases, Department of Clinical and Experimental Medicine, University of Catania, ARNAS Garibaldi Hospital, 95122 Catania, Italy; federicacosentino91@gmail.com (F.C.); serenaspampinato93@gmail.com (S.S.); cacopardobruno@inwind.it (B.C.); giuseppe.nunnari1@unict.it (G.N.); 2Department of Biomedical and Biotechnological Science, Section of Pharmacology, University of Catania, 95123 Catania, Italy; uni365053@studium.unict.it (E.A.); uni318437@studium.unict.it (C.M.B.); gcantare@unict.it (G.C.); bernardi@unict.it (R.B.); 3Clinical Toxicology Unit, University Hospital of Catania, 95123 Catania, Italy; 4Department of Biomedical Sciences, Section of Neuroscience and Clinical Pharmacology, University of Cagliari, 95123 Cagliari, Italy; paola.fadda@unica.it; 5Department of Medicine and Surgery, University of Enna “Kore”, 94100 Enna, Italy; stefano.stracquadanio@unikore.it (S.S.); tnmaniaci29@gmail.com (A.M.); 6Department of Anesthesia and Intensive Care, University Hospital Policlinico “G. Rodolico-San Marco”, 24046 Catania, Italy; luigilavia7@gmail.com

**Keywords:** multidrug-resistant bacteria, antibiotic duration, antimicrobial stewardship, Gram-negative bacterial infections, treatment guidelines, antibiotics PK/PD

## Abstract

Determining the optimal duration of antibiotic therapy for infections caused by multidrug-resistant Gram-negative bacteria (MDR-GNB) is a critical challenge in clinical medicine, balancing therapeutic efficacy against the risks of adverse effects and antimicrobial resistance. This narrative review synthesises current evidence and guidelines regarding antibiotic duration for MDR-GNB infections, emphasising bloodstream infections (BSI), hospital-acquired and ventilator-associated pneumonia (HAP/VAP), complicated urinary tract infections (cUTIs), and intra-abdominal infections (IAIs). Despite robust evidence supporting shorter courses (3–7 days) in uncomplicated infections caused by more susceptible pathogens, data guiding optimal therapy duration for MDR-GNB remain limited, particularly concerning carbapenem-resistant *Enterobacterales* (CRE), difficult-to-treat *Pseudomonas aeruginosa* (DTR-*Pa*), and carbapenem-resistant *Acinetobacter baumannii* (CRAB). Current guidelines from major societies, including IDSA and ESCMID, provide explicit antimicrobial selection advice but notably lack detailed recommendations on the duration of therapy. Existing studies demonstrate non-inferiority of shorter versus longer antibiotic courses in specific clinical contexts but frequently exclude critically ill patients or those infected with non-fermenting MDR pathogens. Individualised duration decisions must integrate clinical response, patient immunologic status, infection severity, source control adequacy, and pharmacologic considerations. Significant knowledge gaps persist, underscoring the urgent need for targeted research, particularly randomised controlled trials assessing optimal antibiotic duration for the most challenging MDR-GNB infections. Clinicians must navigate considerable uncertainty, relying on nuanced judgement and close monitoring to achieve successful outcomes while advancing antimicrobial stewardship goals.

## 1. Introduction

Multidrug-resistant Gram-negative bacteria (MDR-GNBs) pose an increasingly urgent threat to global public health [1]. Of particular concern are the severe infections these organisms cause, such as pneumonia, meningitis, surgical site, and bloodstream infections (BSI), particularly within healthcare environments [2]. Management is becoming increasingly complicated due to the growing resistance of pathogens to a wide range of antimicrobial compounds, including carbapenems, which are often considered a last-resort treatment [3]. Indeed, the presence of resistance can make it difficult or even impossible to effectively treat infections.

MDR-GNB epidemiological burden is consistent [4]: in the United States (US) alone, millions of cases of antimicrobial-resistant infections occur annually, resulting in tens of thousands of deaths and incurring billions of dollars in healthcare costs. The emergence and spread of novel resistance mechanisms further exacerbate this crisis [1]. Notably, the situation has worsened since the outbreak of the SARS-CoV-2 pandemic with data showing that certain hospital-acquired infections caused by resistant bacteria have increased over this period [5]. The prevalence of MDR-GNB varies significantly depending on geographic location, healthcare setting, and patient population. Higher rates are typically observed in intensive care units (ICUs) and long-term care facilities (LTCFs), which can act as reservoirs [6]. Infections caused by MDR-GNB have been consistently linked to adverse clinical outcomes, including increased mortality, prolonged hospital stays, and substantially greater healthcare costs [7,8].

Prior to the advent of the pandemic, concerted efforts to combat infectious diseases had already led to a decline in antimicrobial resistance (AMR)-related mortalities, particularly within the context of hospitals. However, the subsequent period witnessed a disconcerting escalation in nosocomial MDR-GNB infections [5]. Furthermore, the increasing detection of MDR-GNB in community settings complicates the management and control programme [9].

The varied classification of threats by public health organisations (i.e., Centers for Disease Control and Prevention (CDC)’s Urgent/Serious/Concerning lists and World Health Organization (WHO)’s Critical/High/Medium priority pathogens) and the focus of clinical guidelines on specific resistance mechanisms highlight that MDR-GNB are not a uniform entity [10]. This inherent heterogeneity, stemming from diverse bacterial species, differing resistance mechanisms (e.g., enzymatic inactivation, efflux pumps, and target modification), and varying virulence potentials, suggests that a single, universal approach to treatment duration is unlikely to be optimal. The specific pathogen and its resistance profile may significantly influence treatment response and the risk of relapse, consequently affecting the ideal duration of therapy. Nonetheless, generating the pathogen-specific, high-quality evidence needed to inform such nuanced recommendations remains a major challenge [11]. Figure 1 provides an overview of the key problems involved.

## 2. The Duration Dilemma

Deciding on the appropriate length of antibiotic therapy for infections caused by MDR-GNB is a complex medical issue. Clinicians must carefully balance the need for complete bacterial eradication and successful treatment with the significant risks of prolonged drug exposure [12]. Extended antibiotic courses increase the likelihood of drug-related toxicities, ranging from mild adverse reactions to severe organ damage. Furthermore, long-term therapy disrupts patients’ protective microbiome, potentially leading to complications such as *Clostridioides difficile* infection. Continued treatment also contributes to increased healthcare expenditures [13].

Critically, prolonged antibiotic use exerts significant selective pressure, driving the emergence and propagation of further AMR within both the individual patient and the broader healthcare environment [12].

Conversely, prematurely discontinuing antibiotics, especially for severe infections as well as those caused by pathogens known for persistence or virulence, carries the risk of treatment failure, clinical relapse, or developing complications. This balance is particularly precarious when dealing with MDR-GNB, for which therapeutic options may be limited, and the consequences of therapy failing are severe [7,14].

Furthermore, within the MDR-GNB management paradigm, the infections requiring the use of broad-spectrum agents, combination therapies, or drugs with the potential for increased toxicity are precisely the infections for which antimicrobial stewardship principles most strongly advocate limiting antibiotic exposure [15]. The necessity to employ potent, and at times more toxic, regimens to overcome resistance clashes directly with the need to minimise the duration of treatment in order to prevent further resistance development and other adverse consequences [16]. This creates a high-risk clinical decision-making environment, often clouded by the absence of definitive evidence specifically addressing optimal courses for these resistant pathogens in diverse clinical scenarios.

This research aims to provide a synthesis of the current evidence and major guideline recommendations concerning the optimal duration of antibiotic therapy, specifically for infections caused by clinically significant MDR-GNB. It will critically review guidance from leading international societies, analyse findings from comparative clinical studies evaluating shorter versus longer treatment courses across key infection syndromes (e.g., BSI, pneumonia, urinary tract infections, and intra-abdominal infections), discuss pathogen-specific nuances, and delineate factors influencing decision on the duration in contemporary clinical practice. The scope focuses on the common MDR-GNB phenotypes that are encountered in healthcare settings, as well as the relevant evidence for determining appropriate treatment lengths.

## 3. Methods

This narrative review was conducted to summarise the current evidence, expert guidance, and clinical considerations regarding the optimal duration of antibiotic therapy for infections caused by MDR-GNB. A structured, albeit non-systematic, methodology was employed to integrate data from clinical guidelines, peer-reviewed literature, and relevant meta-analyses.

The article focuses on five major infection syndromes: BSI, hospital-acquired and ventilator-associated pneumonia (HAP/VAP), complicated urinary tract infections (cUTIs), intra-abdominal infections (IAIs), and site-specific infections caused by high-priority MDR pathogens (e.g., CRE, DTR-P. aeruginosa, and CRAB). Particular attention was given to studies comparing the duration of treatment (short versus long courses), as well as to pathogen- and host-specific nuances that influence treatment decisions.

A search of English-language articles up to April 2025 was performed using PubMed, Scopus, and Web of Science. Search terms included “antibiotic duration”, “multidrug-resistant”, “Gram-negative”, “carbapenem-resistant”, “ESBL”, “*Pseudomonas aeruginosa*”, “*Acinetobacter baumannii*”, “*Enterobacterales*”, “BSI”, “VAP”, “HAP”, “UTI”, “intra-abdominal infection”, and “short-course therapy”. References cited within articles were also screened to ensure inclusion of pivotal studies, systematic reviews, randomised controlled trials (RCTs), and expert consensus documents. Priority was given to high-quality evidence comprising randomised trials, systematic reviews with meta-analyses, large retrospective cohorts, and updated guidelines from major infectious diseases societies, such as the Infectious Diseases Society of America (IDSA) and the European Society of Clinical Microbiology and Infectious Diseases (ESCMID).

While not designed as a systematic review, the methodology prioritised a balanced and critical appraisal of the available evidence to inform practical clinical decision-making and highlight the key knowledge gaps.

## 4. The MDR-GNB Landscapes

Multidrug-resistant organisms (MDROs) are defined as microorganisms, predominantly bacteria, which are not susceptible to at least one agent in three or more antimicrobial categories. Although some nomenclature refers to resistance to a single agent, e.g., methicillin-resistant *Staphylococcus aureus* (MRSA)—these pathogens frequently exhibit resistance to multiple drug classes [7]. More stringent definitions categorise isolates as extensively drug-resistant (XDR), non-susceptible to all but two or fewer antimicrobial categories, or pan-drug-resistant (PDR), non-susceptible to all agents in all categories [17]. 

Gram-negative bacteria possess structural features, such as an outer membrane, which contribute to intrinsic resistance, and they are also particularly adept at acquiring resistance genes via mobile genetic elements like plasmids, facilitating the rapid spread of resistance determinants [1].

From the perspective of therapeutic intricacy, MDROs groups presenting the most significant challenges are as follows:

**Extended-spectrum β-lactamase-producing *Enterobacterales* (ESBL-E)**, predominantly *Escherichia coli* and *Klebsiella pneumoniae*, produce enzymes (extended-spectrum beta-lactamases, or ESBLs) that hydrolyse and confer resistance to most penicillin and cephalosporin antibiotics, including third-generation agents, as well as aztreonam [18]. Moreover, ESBL-E are at times associated with resistance to other antibiotic classes like fluoroquinolones [19]. Resistance is often mediated by genes like *blaCTX-M-15*, which are carried on plasmids and contribute to the efficient global dissemination of such bacteria in human, animal, and environmental reservoirs [20]. ESBL-E infections, particularly community-acquired ones, are increasing at an alarming rate [9]. 

**Carbapenem-resistant *Enterobacterales* (CRE)** represent a major public health emergency. Resistance is commonly mediated by carbapenemase enzymes, which hydrolyse carbapenems and frequently other β-lactams [21]. Key carbapenemase families include *Klebsiella pneumoniae* carbapenemase (KPC), New Delhi metallo-β-lactamase (NDM), Verona integron-encoded metallo-β-lactamase (VIM), imipenemase metallo-β-lactamase (IMP), and oxacillinase-48-like (OXA-48) enzymes [17]. The CDC has classified CRE as an “urgent threat”, while the WHO has designated it a “critical priority” pathogen [10]. Therapeutic options available are severely restricted, often necessitating combination regimens or newer, costly agents [3].

**Difficult-to-treat resistance *Pseudomonas aeruginosa* (DTR-Pa)**, which, according to the IDSA, refers to isolates that are resistant to piperacillin-tazobactam, ceftazidime, cefepime, aztreonam, meropenem, imipenem-cilastatin, ciprofloxacin, and levofloxacin [22]. Indeed, *P. aeruginosa* is intrinsically resistant to many antibiotics and readily acquires further resistance mechanisms [1]. DTR-*Pa* poses significant treatment challenges, particularly when dealing with HAP/VAP [23]. WHO classifies carbapenem-resistant *P. aeruginosa* (CRPA) as a “high priority” pathogen [24].

**Carbapenem-resistant *Acinetobacter baumannii* (CRAB)** causes infections frequently encountered in ICUs that are associated with high mortality rates, especially HAP/VAP as well as BSI [25]. Indeed, *A. baumannii* is an opportunistic pathogen notorious for its ability to acquire resistance to virtually all classes of antibiotics, including carbapenems [26]. CRAB is designated an “urgent threat” by the CDC and a “critical priority” by the WHO [10]. 

**AmpC β-lactamase producers**: Certain *Enterobacterales* (e.g., *Enterobacter* spp., *Citrobacter* spp., and *Serratia* spp.) possess chromosomal or plasmid-mediated AmpC β-lactamases. These enzymes confer resistance to penicillins, extended-spectrum cephalosporins (with the exception of cefepime in some cases), cephamycins (e.g., cefoxitin), and β-lactam/β-lactamase inhibitor combinations (e.g., amoxicillin-clavulanate and piperacillin-tazobactam) [27,28].

***Stenotrophomonas maltophilia*** is a non-fermenting Gram-negative bacillus (NF-GNB) exhibiting intrinsic resistance to numerous antibiotics, most notably carbapenems. It represents an important opportunistic pathogen, particularly in hospitalised or immunocompromised patients [7] [28] with difficult to approach therapeutic choices [29].

## 5. Guideline Perspectives

Major infectious diseases societies provide guidance on the management of AMR infections with a primary focus on the selection of appropriate antimicrobials. However, specific recommendations regarding the duration of therapy, particularly for MDR-GNB, are often limited.

### 5.1. IDSA AMR Guidance (Versions 2020–2024)

The IDSA has published iterative guidance documents focusing on the treatment of infections caused by ESBL-E, CRE, DTR-Pa, CRAB, AmpC-producing *Enterobacterales*, and *S. maltophilia*. Developed by expert panels, these documents are based on comprehensive literature reviews (though not necessarily systematic reviews using Grading of Recommendations Assessment, Development, and Evaluation (GRADE) methodology). They primarily aim to help clinicians select effective antibiotic regimens, including newer agents and combination strategies [30].

A crucial tenet explicitly stated in the IDSA guidance is that prolonged treatment courses are not necessarily required for infections caused by AMR pathogens compared to those caused by more susceptible bacteria. The emphasis is on adequately treating the infection based on clinical parameters, rather than extending the duration solely due to the resistance phenotype [31].

Notably, the guidance advises that duration should adhere to standard clinical principles and be individualised based on several aspects regarding both patients and infection specifics. Host-related factors, namely patient immune status, and overall ability to combat infection, as well as subsequent general response to initiated antibiotic therapy, must be considered [32]. In terms of the particular features of the infection, it is vital to gain effective and prompt control over the source (e.g., draining of abscesses, extraction of infected devices) [33]. Furthermore, the site and severity are implicitly considered, as the durations required for different types of infections (e.g., uncomplicated cystitis vs. pyelonephritis/cUTIs) generally differ [34].

The guidance offers some context-specific advice, as several clinical scenarios tend to recur. For uncomplicated cystitis in particular, if a patient improves clinically despite receiving empiric therapy that is later found to be inactive against the cultured organism, it may not be necessary to change the antibiotics or extend the planned treatment duration. However, for more severe infections (i.e., anything other than uncomplicated cystitis), if the initial empiric therapy was potentially inactive based on the results of susceptibility testing, it is recommended that a change be made to an active regimen for a full treatment course, timed from the initiation of the active agent [23]. For cUTIs, durations analogous to those observed in pyelonephritis are recommended. However, if the underlying cause is effectively managed and any complicating factors are addressed, shorter durations, similar to those seen in uncomplicated cystitis may be considered appropriate [23].

Updates and consultations are periodically made available. Indeed, the version v4.0 of December 2023 refined the dosing recommendations for specific agents (e.g., high-dose ampicillin-sulbactam for CRAB ceftazidime-avibactam plus aztreonam for MBL-CRE and *S. maltophilia*) [35], and updated preferred treatment options (e.g., favouring sulbactam-durlobactam for CRAB) [36]. However, no specific duration mandates have been introduced. The guidance consistently recommends consulting infectious disease specialists when managing these complex conditions [22].

### 5.2. ESCMID Guidelines (2021)

The 2021 ESCMID guidelines, endorsed by the European Society of Intensive Care Medicine (ESICM), focused on targeted antibiotic treatment for infections caused by third-generation cephalosporin-resistant *Enterobacterales* (3GCephRE, conceptually similar to ESBL-E) and carbapenem-resistant Gram-negative bacteria (CRE, CRPA, and CRAB) [37]. These guidelines employed the GRADE methodology based on systematic literature reviews [38].

The primary emphasis was placed on the evaluation of the comparative effectiveness of different antibiotic agents. In addition, the role of combination therapy versus monotherapy for specific resistant pathogen groups was investigated [39,40]. Importantly, these guidelines do not offer specific recommendations on the duration of antibiotic therapy for MDR-GNB infections.

The main outcome assessed in their evidence review was all-cause mortality, typically after 30 days [37]. However, this reflects the timeframe for assessing outcomes rather than the recommended length of treatment.

The guideline panel highlighted the limitations of the available evidence, noting that many comparisons relied on small observational studies prone to bias. Evidence regarding the efficacy of newer β-lactam/β-lactamase inhibitors against carbapenem-resistant organisms was particularly scarce at the time, meaning that most recommendations were based on very low- or low-certainty evidence [41].

ESCMID is currently updating the 2021 guidelines, with an anticipated completion date of April 2026. This ongoing process is currently reported to be in the Population, Intervention, Comparison, and Outcome (PICO) question generation stage [42,43].

### 5.3. Comparison and Other Guidelines

Differences exist between major guidelines in terms of methodology and specific recommendations, thus potentially leading to diverse recommendations, such as the contrasting views on the role of piperacillin-tazobactam for non-severe ESBL-E infections [22,44].

Guidelines from other societies also exist, such as the Spanish Society of Infectious Diseases and Clinical Microbiology (SEIMC), often combining systematic reviews with expert opinion [27]. Table 1 presents a comparison of the general treatment approaches (agent selection and combination therapy) recommended by IDSA and ESCMID for key MDR-GNB.

The conspicuous absence of explicit, evidence-based recommendations on treatment duration within major guidelines, particularly the 2021 ESCMID document [37], stands in contrast to the detailed advice provided on antimicrobial agent selection. This is likely indicative of two interconnected factors: Firstly, a paucity of high-quality clinical trial data specifically designed to compare different treatment durations for infections caused by various MDR-GNB across different clinical syndromes [15]. Secondly, a potentially consensus view, explicitly articulated by IDSA [23], that duration should primarily be dictated by general clinical principles that apply similarly to both susceptible and resistant infections. Irrespective of the underlying rationale, this absence of precise guidance imposes a substantial interpretive burden on clinicians tasked with the management of complex MDR-GNB infections. Indeed, they must extrapolate from general principles or from studies that often excluded the patient populations and resistant pathogens they are treating, navigating considerable uncertainty when determining the optimal, shortest effective course of therapy [45].

## 6. Evidence Review: Short Versus Long Antibiotic Courses

A substantial body of literature has investigated the comparative effectiveness and safety of shorter versus longer durations of antibiotic therapy for a wide range of bacterial infections [13,46]. A primary motivation for these studies is the promotion of antimicrobial stewardship [47]. Reducing unnecessary antibiotic exposure is a cornerstone strategy for mitigating the selection pressure driving AMR, minimising the risk of antibiotic-associated adverse reactions (including *C. difficile* infection) [48,49], reducing disruption to the host microbiome [50], and potentially lowering healthcare costs [51,52].

Across many common bacterial infections frequently encountered in both community and hospital settings, such as community-acquired pneumonia (CAP), acute bacterial sinusitis, uncomplicated cellulitis, uncomplicated UTIs in women, and IAIs with adequate source control, the evidence robustly supports the use of shorter antibiotic courses [15].

Consistent trial results show that shorter treatment periods (typically 3–7 days, depending on the infection) are as effective as longer courses (e.g., 7–14 days or more) in terms of achieving clinical cure and preventing short-term mortality. Relapse rates are also often comparable, although nuances exist for specific scenarios [13]. One systematic review found that 85% of 315 trials comparing short versus long courses concluded non-inferiority or equivalence for clinical outcomes [15]. Extensive reviews compiling RCTs further support the “shorter is better” approach for a wide array of infections, including various types of pneumonia (CAP and VAP), UTIs, skin infections, osteomyelitis, septic arthritis, meningitis, and others, often demonstrating non-inferiority of shorter courses (ranging from single doses to 7 days or several weeks, depending on the infection) compared to traditionally longer regimens [53].

However, a key issue that permeates much of this research is the frequent exclusion of patients with the most complex or severe infections. Many trials comparing antibiotic durations have specifically excluded patients who are critically ill, severely immunocompromised, infected with highly resistant pathogens (including specific MDR-GNB, such as *P. aeruginosa* or *Acinetobacter*), or those in whom adequate source control could not be achieved. This significantly limits the direct generalisability of findings supporting shorter durations to the management of severe infections caused by MDR-GNB, which is the very focus of this report [45].

Significant investigation has been conducted in the area of BSI, particularly regarding cases caused by Gram-negative microorganisms. A substantial body of research, encompassing multiple RCTs and subsequent meta-analyses, including large-scale individual participant data (IPD) ones, provides robust evidence that for uncomplicated Gram-negative BSI, a 7-day course of antibiotic therapy is non-inferior to a 14-day course [54,55,56]. This conclusion holds for key outcomes including 30-day and 90-day all-cause mortality rates, recurrence of bacteraemia (relapse), length of hospital stays, readmission rates, and adverse events [57]. The recent large-scale BALANCE trial, encompassing over 3700 patients, confirmed the high probability of non-inferiority for a 7-day versus a 14-day treatment period with regard to 90-day mortality outcome [58]. The findings outlined here pertain predominantly to bacteraemia caused by *Enterobacterales* (the vast majority of isolates in these trials), which frequently originate from a urinary source, in patients who achieve clinical stability (e.g., afebrile for 48 h) and have adequate source control.

Despite the robustness of these findings for uncomplicated cases, significant caveats remain. The trials frequently exhibited elevated screening failure rates suggesting that the included patients may not fully represent the broader population with Gram-negative BSI.

Critically, patients with infections caused by MDR pathogens, particularly NF-GNB such as *P. aeruginosa* and *A. baumannii*, were significantly underrepresented or excluded. Similarly, patients with severe immunosuppression, persistent bacteraemia, endovascular infection sources, or uncontrolled foci of infection were typically not included [45]. Furthermore, the BALANCE trial noted substantial protocol deviations, with many physicians opting for longer durations than assigned, suggesting clinician reluctance or scepticism regarding shorter courses, particularly in perceived higher-risk scenarios [58]. Consequently, while the 7-day duration is adequately substantiated for uncomplicated *Enterobacterales* bacteraemia, direct extrapolation to complex BSI involving MDR pathogens, non-fermenters, or unstable patients demands considerable caution [59]. A retrospective study was conducted in order to compare short (≤7 days) with long (>7 days) therapy specifically for *P. aeruginosa* or *Acinetobacter* BSI and found no difference in 30-day mortality, even in the carbapenem-resistant subgroup. However, it requires prospective validation [60].

For HAP/VAP, international guidelines, such as those from the IDSA/American Thoracic Society (ATS) and European societies, generally recommend a 7-day course of antibiotic therapy [61]. This recommendation is based on several RCTs and meta-analyses demonstrating that an 8-day (or 7-day) course is non-inferior to longer courses (typically 10–15 days) in terms of clinical cure rates and mortality. Shorter treatment durations are associated with a significantly higher number of antibiotic-free days, which is a key stewardship outcome [62].

A crucial and persistent caveat relates to VAP caused by NF-GNB, specifically *P. aeruginosa* and *Acinetobacter* spp. Several studies, including subgroup analyses of initial trials and subsequent meta-analyses, have reported a higher risk of pulmonary infection recurrence or relapse when treating these specific pathogens with shorter (7–8 day) courses compared to longer regimens. Although this increased recurrence has not invariably resulted in poorer patient outcomes or prolonged ICU stays in meta-analyses, it remains a significant concern for clinicians managing these difficult-to-treat pathogens [62,63]. Consequently, the optimal duration for VAP caused by MDR *P. aeruginosa* or CRAB remains the subject of active debate, with clinical practice often favouring longer durations (e.g., ≥14 days) despite the lack of definitive evidence showing benefit [64]. A recent retrospective study focusing specifically on MDR *P. aeruginosa* HAP/VAP found no improvement in clinical success or mortality with treatment beyond 8 days, adding complexity to the issue [65].

For acute pyelonephritis caused by *Enterobacterales*, including ESBL-producing strains, a 7-day course of appropriate antibiotics is widely regarded as adequate for patients exhibiting a favourable clinical response [64]. Evidence suggests that this duration is non-inferior to longer courses (e.g., 10–14 days). One trial specifically conducted on afebrile men with UTI (a form of cUTI) found that a 7-day treatment with either ciprofloxacin or trimethoprim-sulfamethoxazole (TMP-SMX) was non-inferior to 14 days for resolving symptoms [66]. A retrospective study focusing on cUTIs caused by ESBL-E reported no significant difference in 30-day mortality or a composite outcome of mortality/reinfection between patients receiving ≤7 days versus >7 days of therapy [67]. Furthermore, a large observational study of hospitalised patients with cUTI and associated bacteraemia suggested that a 7-day course was as effective as 14 days (and potentially better than 10 days) in preventing recurrence, provided patients received intravenous (IV) β-lactams throughout or were transitioned to oral agents with high bioavailability [68].

Nevertheless, evidence supporting shorter courses is less robust for cUTI caused by *P. aeruginosa*. Clinical practice guidelines and expert opinions often recommend longer durations, typically 10–14 days, for pseudomonal pyelonephritis, particularly if a slow response or complications are present. Data on the optimal duration of treatment for UTIs caused by CRE or DTR-*P. aeruginosa* are very limited. It is also important to distinguish between pyelonephritis/cUTI and prostatitis, as shorter courses are associated with higher relapse rates, necessitating longer treatment (typically ≥ 14 days). For catheter associated UTIs, durations of 3–7 days may be appropriate depending on clinical response and catheter removal [64].

In complicated IAIs where adequate source control has been achieved (i.e., surgical drainage, resection, or repair), robust evidence supports short, fixed courses of postoperative antibiotic therapy [69,70]. The landmark STOP-IT trial randomised patients with cIAI and adequate source control to receive either antibiotics until 2 days after clinical resolution (median duration 8 days) or a fixed 4-day course. The trial found no significant difference in the composite outcome of surgical site infection, recurrent IAI, or death between the two groups [71]. Based on such supporting data, major surgical and infectious disease guidelines now generally recommend short courses of antibiotics (typically 3–5 or 4–7 days) following definitive source control for cIAI [72]. A retrospective study of surgical ICU patients with BSI secondary to IAI also found that stopping antibiotics within 7 days of source control did not increase recurrence [73].

The STOP-IT trial and much of the substantiating evidence primarily involved patients with community-acquired IAIs and did not specifically focus on infections caused by MDR-GNB. Data guiding optimal duration for IAIs known or suspected to be caused by CRE, DTR-P. aeruginosa, or CRAB, particularly in critically ill patients, those with healthcare-associated infections, or situations where source control is delayed or incomplete, remain scarce. In such complex scenarios, clinicians may still consider longer durations (e.g., ≥7–14 days), although the evidence base for this practice is limited and largely extrapolated or based on expert opinion [72]. The fundamental principle remains that adequate source control is paramount; without it, duration of antibiotic therapy is secondary to achieving definitive control of the infectious focus [74]. Table 2 summarises the literature evidence regarding antibiotic duration.

The collective findings reveal a distinct pattern: robust support exists for shortening antibiotic durations for many common, relatively uncomplicated Gram-negative infections, particularly those caused by more susceptible organisms and where source control has been definitively achieved [15]. However, the strength of the evidence base remarkably diminishes when considering infections caused by highly resistant MDR pathogens, namely CRE, DTR-P. aeruginosa, and CRAB, severely ill or immunocompromised patients, or clinical situations where source control is uncertain or incomplete [45].

This creates a two-tiered evidence landscape: in easier clinical scenarios, the path towards shorter, guideline-supported durations is clearer. For complex MDR-GNB infections, which pose the greatest challenges, evidence supporting specific durations is often inconclusive, contradictory (e.g., NF-GNB VAP recurrence [62]), or non-existent. Consequently, management decisions in such tricky cases are heavily reliant upon extrapolation from potentially inapplicable data, expert opinion, and individualised clinical judgement, leading to considerable variability in practice and uncertainty regarding the very optimal duration.

## 7. Nuances in Duration Decisions for Specific MDR Pathogens

While the general principle of avoiding unnecessarily long antibiotic courses applies broadly, the specific characteristics of certain high-priority MDR-GNB and the limited evidence available brings about important nuances into duration decisions.

For infections caused by ESBL-E, the extant evidence generally supports adherence to standard treatment durations appropriate for the specific infection site, assuming that an effective agent is used [86]. Subgroup analyses within large BSI trials did not show significantly different outcomes between short (7-day) and long (14-day) courses for patients with ESBL-E bacteraemia, although these analyses may have been underpowered [45,87].

Studies focusing on ESBL-E UTIs also suggest that shorter courses (e.g., ≤7 days for complicated UTI) can be effective without increasing mortality or short-term recurrence [67]. This aligns with the IDSA principle that resistance phenotype alone should not mandate an extended therapeutic regimen. A pragmatic consideration is that plasmids carrying ESBL genes often harbour resistance determinants to other antibiotic classes (i.e., fluoroquinolones or aminoglycosides) [88], which primarily impacts the therapy choice and could indirectly influence duration if perceived treatment complexity leads clinicians towards longer courses. It is reasonable to conclude that standard durations, such as 7 days for uncomplicated BSI or pyelonephritis, appear appropriate based on current data [86].

There is a marked scarcity of high-quality clinical trial data specifically comparing different treatment durations for infections caused by CRE [89]. The majority of literature focuses on the identification of effective antimicrobial regimens, often involving combinations of older agents (polymyxins, tigecycline, aminoglycosides, and high-dose carbapenems), newer β-lactam/β-lactamase inhibitors (ceftazidime-avibactam, meropenem-vaborbactam, and imipenem-relebactam) [90], or cefiderocol [3,91]. Given the high mortality associated with CRE infections, the limited therapeutic armamentarium, and the complexity of treatment regimens, clinicians may intuitively favour longer durations to ensure eradication, despite the lack of explicit guideline mandates or supporting evidence for this practice [92]. Duration decisions for CRE infections are currently driven more by clinical response, source control, and severity, rather than by evidence-based duration comparisons.

Managing infections caused by DTR-P. aeruginosa presents substantial hurdles. While IDSA provides recommendations for preferred agents (e.g., ceftolozane-tazobactam, ceftazidime-avibactam, imipenem-relebactam, and cefiderocol), it offers no specific duration guidance [23]. The prevailing concern regarding the elevated recurrence rates associated with short-course therapy for P. aeruginosa VAP exerts a significant influence on clinical practice [62]. Consequently, longer durations (e.g., 10–14 days or more) are frequently employed for severe *P. aeruginosa* infections like pneumonia or pyelonephritis [64]. However, the retrospective finding that treatment beyond 8 days offered no benefit for MDR *P. aeruginosa* HAP/VAP challenges this assumption and underscores the uncertainty [65]. The duration of treatment for DTR-P. aeruginosa infections is highly individualised, with a need to balance stewardship with the perceived risk of relapse.

Treating CRAB infections represents one of the most significant challenges in infectious diseases due to extreme resistance profiles [93,94]. The most effective treatment regimens generally involve combination therapy, frequently comprising high-dose ampicillin-sulbactam or newer agents such as sulbactam-durlobactam or cefiderocol, in conjunction with polymyxins, tigecycline, or minocycline [26,95]. The evidence guiding optimal treatment duration is exceptionally limited. Clinical trials and practice often involve treatment courses ranging from 7 to 21 days, frequently extending to 14 days or longer, particularly for severe infections like VAP or bacteraemia [96]. A study of cancer patients treated with colistin for CRAB infections reported higher mortality in those receiving treatment for <14 days than for ≥14 days. However, this finding is specific to a vulnerable population and an agent with complex pharmacokinetics and toxicity [97]. Distinguishing true infection from colonisation is of critical importance in the management of CRAB isolates, in order to avoid unnecessary, potentially toxic, and resistance-promoting therapy [98].

*Pseudomonas* and *Acinetobacter*, the non-fermenter pathogens, are central to the debate surrounding the safety of short-course antibiotic therapy. In comparison to *Enterobacterales*, they possess different intrinsic virulence factors, a greater propensity for biofilm formation, and often exhibit more complex resistance mechanisms [60]. These biological differences may contribute to the observations of higher recurrence rates with shorter treatment durations in some VAP studies [62]. This finding suggests that the biology of pathogens may interact significantly with the duration of treatment, thereby necessitating greater caution when using short-course strategies, which have been developed primarily from studies conducted on *Enterobacterales*, in infections caused by these non-fermenters, particularly pneumonia.

Despite mounting evidence supporting shorter courses in specific contexts and guideline principles advising against extending duration solely for resistance, clinicians often opt for longer treatment regimens in practice, particularly for MDR-GNB infections. This tendency is attributable to a number of factors, including the severity of clinical presentation, host immunosuppression, difficult-to-treat pathogens, and uncertainty regarding adequate source control [45]. Concerns also regard the potential for treatment failure, relapses, or recurrences, especially when limited salvage therapy options are available. Ultimately, key obstacles that remain to be overcome are the lack of definitive, high-quality data supporting shorter durations in specific complex clinical scenarios and ingrained professional habits or scepticism regarding the safety and efficacy of shorter courses for severe infections [15,99]. This inevitably leads to considerable variability in practice and highlights the urgent need for targeted research to optimise treatment duration for these highest-priority MDR pathogens, while fostering both efficacy and responsible stewardship.

## 8. Key Factors Guiding Individualised Therapy Duration

Clinical decision-making must rely on a careful, individualised assessment integrating multiple patient-, infection-, and treatment-related factors.

The patient’s clinical response is arguably the most critical factor guiding the duration of treatment. Ongoing assessment—including resolution of fever, normalisation of white blood cell count, improvement in signs and symptoms specific to the infection site, and stabilisation or improvement of organ function—is paramount. Studies supporting shorter durations often used criteria such as being afebrile and haemodynamically stable for at least 48 h prior to potential discontinuation [100]. A slow or inadequate clinical response should prompt a thorough re-evaluation of the diagnosis, the adequacy of source control, and the appropriateness of the chosen antibiotic regimen and its duration [101].

While evidence is evolving, caution is generally warranted in critically ill patients presenting with severe sepsis/septic shock requiring ICU admission or who are significantly immunocompromised. Indeed, longer durations may be considered, although prolonged therapy also carries risks in this population [102]. Severity scores like APACHE II or SOFA may help stratify risk, but do not definitively dictate duration.

The overall assessment should encompass any underlying medical conditions that could influence both the patient’s response to infection and the pharmacokinetics/pharmacodynamics (PK/PD) of antibiotics.

With regard to the specifics of infection, the anatomical location is a primary determinant of the standard treatment duration. Uncomplicated cystitis usually requires 3–5 days of treatment, whereas pyelonephritis generally warrants at least 7 days [64]. Pneumonia guidelines recommend a duration of 7 days for HAP/VAP, but longer courses may be considered for NF-GNB [80]. Deeper, more complex infections like endocarditis, osteomyelitis, meningitis, and undrained abscesses inherently require significantly longer treatment courses, often spanning weeks [103]. Moreover, achieving adequate and timely source control is fundamental, particularly for infections like IAI, catheter-related BSI, and those associated with prosthetic material [22]. In cases where the source of infection is not successfully contained, prolonged antibiotic therapy is often required, frequently necessitating further intervention [85]. Most evidence supporting shorter antibiotic courses explicitly assumes or requires adequate source control as a prerequisite [15]. The specific agent involved may also be relevant, as the inherent virulence and the potential for complications associated with certain pathogens might subtly influence clinician decisions on duration [104].

Furthermore, pharmacological considerations must be taken into account during the decision-making process. The chosen oral agent must demonstrate reliable bioavailability, achieve adequate concentrations at the site of infection, and demonstrate in vitro activity against the specific MDR pathogen. The ability to transition from IV to an appropriate oral antibiotic regimen can facilitate earlier hospital discharge but does not inherently shorten the total duration of effective therapy required [105,106]. A study on cUTI with bacteraemia suggested that 7-day courses were effective when patients received continuous IV β-lactams or were switched to oral agents with excellent bioavailability (fluoroquinolones and TMP-SMX), whereas 10 days might be necessary if transitioning to agents with lower or less reliable bioavailability [68]. Employing PK/PD principles to optimise antibiotic posology, such as administering higher doses or extended/continuous infusions for time-dependent agents—like β-lactams—aims to maximise bacterial killing and improve clinical outcomes [28]. While the strategies appear to be intuitively appealing, further dedicated study is required to determine their direct impact on enabling shorter treatment durations. However, achieving a more rapid clinical response could potentially support earlier discontinuation.

The strong emphasis placed on dynamic clinical assessment (response to therapy and resolution of signs/symptoms), the critical importance of source control, and the need to consider individual host factors, particularly in the context of limited definitive duration evidence for complex MDR-GNB infections, signifies a necessary shift in the decision-making paradigm. Rather than adhering rigidly to predetermined, definite durations (e.g., “always treat CRE for 14 days”), a more fluid and individualised approach is taken for these challenging infections. The focus is on identifying the earliest point at which therapy can be safely discontinued based on the patient’s demonstrated clinical improvement and control of the infectious process. This necessitates vigilant monitoring, strong clinical judgement, and potentially the adjunctive use of biomarkers or repeated assessments to guide therapy cessation, moving away from fixed protocols.

Role of Pharmacokinetics/Pharmacodynamics (PK/PD), Biomarkers, and Therapeutic Drug Monitoring (TDM)

Beyond standard clinical assessment, leveraging pharmacological and biological tools can support more nuanced duration decisions. The significant pharmacokinetic/pharmacodynamic (PK/PD) variability observed in critically ill patients complicates standard dosing, potentially leading to sub-therapeutic exposures or toxicity, both of which can compromise outcomes [107,108].

Employing PK/PD principles to optimise antibiotic posology—such as administering higher doses or using extended/continuous infusions for time-dependent agents like β-lactams—aims to maximise bacterial killing and improve clinical outcomes. While the primary goal of such strategies is to enhance efficacy, achieving a more rapid and robust clinical response could, in turn, provide greater confidence for discontinuing therapy earlier [109,110,111]. 

In this context, Therapeutic Drug Monitoring (TDM) emerges as a critical tool for individualization. TDM involves measuring antibiotic concentrations in a patient’s blood to ensure that dosing is adequate to be effective against the pathogen while minimising the risk of dose-dependent toxicity. This is especially crucial for antibiotics with narrow therapeutic indices, such as aminoglycosides and polymyxins, which are often required for treating CRE and CRAB infections. By using TDM to tailor doses to each patient’s unique physiology, clinicians can more confidently achieve target exposures, potentially leading to better and faster clinical responses that could support a shorter overall duration of therapy [112,113,114].

Additionally, biomarkers such as procalcitonin (PCT) and C-reactive protein (CRP) have been investigated to guide the discontinuation of antibiotics. Multiple meta-analyses have concluded that using PCT-guided algorithms can safely reduce the total duration of antibiotic therapy in critically ill patients with some analyses also suggesting a mortality benefit. However, the quality of earlier evidence has been debated, limiting confidence in widespread adoption [115].

The most robust recent evidence comes from the large-scale ADAPT-Sepsis trial, which provided a head-to-head comparison in critically ill patients with sepsis. The trial found that a daily PCT-guided protocol led to a modest but statistically significant reduction in the total duration of antibiotic treatment compared to standard care (a mean difference of approximately one day). Importantly, this was achieved without compromising safety, as the PCT protocol was found to be non-inferior regarding 28-day mortality. In contrast, a daily CRP-guided protocol did not significantly reduce antibiotic duration compared to standard care [116].

Despite these positive findings for PCT, it is important to note that the absolute reduction in antibiotic use was modest, and clinician adherence to the biomarker-guided “stop” recommendations was imperfect. This suggests that while these tools are valuable, their real-world impact depends heavily on their integration into multifaceted antimicrobial stewardship strategies that address barriers to clinician trust and implementation. The precise role of these biomarkers in the specific context of infections caused by high-priority MDR-GNB remains an area for future investigation.

## 9. Knowledge Gaps and Future Directions

Notwithstanding the advances that have been made, there are still significant knowledge gaps with regard to MDR-GNB infections. It is imperative that these gaps are addressed in order to enhance outcomes and optimise antibiotic stewardship. 

High-quality RCTs, specifically evaluating optimal antibiotic durations, are urgently needed in populations frequently excluded from previous trials. Indeed, data from ICU settings are limited, yet critically ill patients often harbour MDR pathogens and have complex physiology impacting treatment [45,102]. Trials like BALANCE are beginning to address this, but more research is required [99]. Similarly, patients with neutropenia, solid organ or hematopoietic stem cell transplants, or receiving immunosuppressive therapies are generally excluded, despite facing high risks from both infection and prolonged antibiotic exposure [117]. While some reports exist, evidence to guide treatment duration in children is less extensive than in adults, even though they represent an exceptionally vulnerable population [118].

The burden of AMR is often highest in low- and middle-income countries (LMICs), yet the vast majority of duration trials are conducted in high-income settings [15]. Research applicable to resource-limited settings is essential.

The most significant gap lies in duration data for infections caused by the highest-priority MDR pathogens: CRE, DTR-*P. aeruginosa*, and CRAB [119]. Existing trials are often underpowered for subgroup analyses of these organisms. Future trials should ideally stratify or focus specifically on these pathogens and compare durations of currently recommended (often newer) therapies.

Although biomarkers such as procalcitonin (PCT) and C-reactive protein (CRP) have been investigated for use in guiding the discontinuation of antibiotics, their precise role and utility, particularly in cases of severe infection, in immunocompromised patients, and in patients with MDR-GNB, require further validation through robust clinical trials [120,121]. Standardised protocols and thresholds need refinement.

The increasing availability of rapid diagnostic tests (RDTs), including molecular methods identifying pathogens and resistance genes directly from specimens, holds promise for enabling earlier, targeted therapy [122,123,124,125,126,127]. Further research is needed to determine the most effective integration of RDTs into clinical workflows, with the potential to facilitate shorter and more precise antibiotic durations.

Most duration trials focus on relatively short-term outcomes (e.g., 30- or 90-day mortality or clinical cure at the end of therapy or during short-term follow-up). More research is needed to understand the impact of different treatment durations on longer-term outcomes, such as relapse rates beyond 90 days, functional status, quality of life, and, crucially, the subsequent emergence of antibiotic resistance within the individual patient’s microbiome.

The concentration of these knowledge gaps within the most complex and high-risk clinical scenarios is very significant. Although the principles of antimicrobial stewardship strongly advocate for shorter durations across the board, the solid evidence base currently available primarily supports this approach in relatively lower-risk situations.

The effective bridging of this evidence gap necessitates the implementation of dedicated clinical trials that are meticulously designed to address these challenging contexts. Nevertheless, conducting such trials presents considerable ethical and logistical hurdles due to patient heterogeneity, the severity of illness, and the difficulty in controlling confounding variables. Until such evidence emerges, clinicians managing these high-risk MDR-GNB infections will continue to operate in a zone of significant uncertainty, potentially leading to overly cautious (and unnecessarily prolonged) treatment courses that may inadvertently contribute to the very resistance problem they aim to combat.

## 10. Conclusions and Recommendations

The practical implication for physicians managing MDR-GNB infections is nuanced. Consultation with infectious diseases specialists or clinical pharmacists with expertise in AMR is strongly recommended to assist with challenging decisions regarding appropriate antimicrobial selection, dosing, and, critically, the duration of therapy.

While the overarching philosophy of “shorter is often better” [128] is driven by stewardship principles and supported by evidence in less complex situations, applying this rigidly to severe infections caused by highly resistant pathogens is not currently supported by robust data. The decision requires a meticulous, individualised risk-benefit analysis for each patient.

There is no universally “correct” duration for many MDR-GNB infections. The goal is to identify the shortest duration that confidently ensures clinical resolution and minimises relapse risk for a specific patient and their specific infection. Achieving this requires a holistic and adaptive approach that combines vigilant clinical assessment, timely source control, and the judicious use of tools like TDM and biomarkers. This underscores the indispensable role of clinical expertise and collaborative decision-making in navigating this challenging therapeutic landscape, while highlighting that further research specifically targeting these evidence gaps remains imperative to provide clinicians with more definitive guidance.

## Figures and Tables

**Figure 1 ijms-26-06905-f001:**
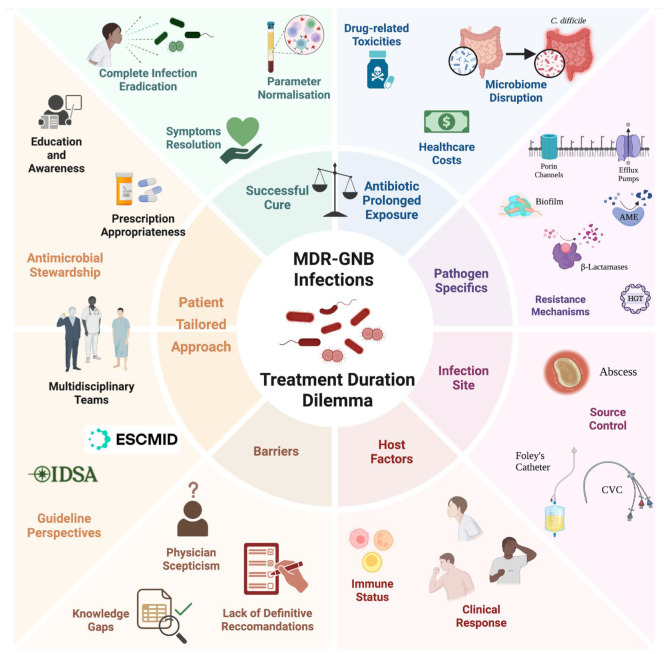
Core issues related to antibiotic therapy duration for MDR Bacterial Infections. Infections caused by multidrug-resistant Gram-negative bacilli (MDR-GNB) inevitably raise the question of the optimal duration of treatment. The choice of antibiotic and dosage regimen is influenced by factors intrinsic to the pathogen, such as resistance mechanisms and virulence, as well as the site of infection and host characteristics. Furthermore, antimicrobial stewardship must not be ignored or deviated from. Therefore, it is imperative to establish individualised therapy for each patient, carefully balancing the need for complete bacterial eradication and successful treatment with the significant risks of prolonged drug exposure. Created with BioRender.com, https://www.biorender.com/ (accessed 17 June 2025). AME—aminoglycoside-modifying enzyme; HGT—horizontal gene transfer; CVC—central venous catheter.

**Table 1 ijms-26-06905-t001:** Comparison of IDSA (v4.0, 2023) and ESCMID (2021) guideline approaches to agent selection for key MDR-GNB infections.

Pathogen/ Resistance Type	IDSA Approach	ESCMID Approach	Key Differences/Notes
**ESBL-E/3GCephRE**	*Preferred:* Carbapenems (ertapenem for cystitis/mild; meropenem/imipenem for others). *Alternatives*: Consider fluoroquinolones, TMP-SMX, nitrofurantoin, and fosfomycin based on susceptibility and site (esp. UTI). Piperacillin-tazobactam non-inferiority data debated, potentially less reliable for severe BSI.	*Severe Infection*: Carbapenem (imipenem/meropenem recommended; ertapenem possible for BSI without shock). *Low-risk/Non-severe:* Suggests piperacillin-tazobactam, amoxicillin-clavulanate, or quinolone. TMP-SMX is considered for non-severe cUTI.	ESCMID more permissive of β-lactam/β-lactamase inhibitors for non-severe infections than IDSA, which emphasises carbapenems more broadly. Neither provides specific duration guidance [23].
**CRE**	*Preferred (KPC/OXA-48):* Ceftazidime-avibactam, meropenem-vaborbactam, and imipenem-relebactam. *Preferred (MBL):* Ceftazidime-avibactam + aztreonam, and cefiderocol. *Alternatives (UTI/IAI)*: Aminoglycosides, cefiderocol, tigecycline/eravacycline, and polymyxins (cystitis only) based on susceptibility/site.	*Severe (KPC/OXA-48)*: Meropenem-vaborbactam or ceftazidime-avibactam if active. *Severe (MBL/Resistant to others)*: Conditionally recommends cefiderocol. Suggests ceftazidime-avibactam + aztreonam. *Non-severe *: Consider older agents if active (e.g., aminoglycosides for cUTI). Combination therapy is suggested if susceptible only to older agents or new agents are unavailable.	General alignment on newer agents for severe infections. IDSA provides more tiered options. Neither provides specific duration guidance [23]. ESCMID used GRADE [37].
**DTR-P. aeruginosa/CRPA**	*Preferred*: Ceftolozane-tazobactam, ceftazidime-avibactam, and imipenem-relebactam. Cefiderocol an alternative. Aminoglycosides (amikacin/tobramycin) alternative for UTI. High-dose extended-infusion traditional β-lactams (e.g., cefepime) if susceptible. Cefiderocol is preferred for MBL strains.	*Severe:* Suggests ceftolozane-tazobactam if active. Cefiderocol is suggested for cUTI. *Non-severe:* Use older active agents. Combination therapy (two active drugs) is suggested if relying on polymyxins, aminoglycosides, or fosfomycin for severe infection.	IDSA provides more options with newer agents. ESCMID emphasises ceftolozane-tazobactam. Neither provides specific duration guidance [23]. WHO moved CRPA from Critical to High Priority [24].
**CRAB**	*Preferred:* Sulbactam-durlobactam (+carbapenem background). *Alternative*: High-dose ampicillin-sulbactam (e.g., 27 g/day) + ≥ 1 other active agent (e.g., polymyxin, minocycline, tigecycline/eravacycline, and cefiderocol).	*HAP/VAP (Sulbactam-S):* Suggests ampicillin-sulbactam. *Sulbactam-R:* Suggests polymyxin or high-dose tigecycline if active. Cefiderocol, polymyxin-meropenem, and polymyxin-rifampin combinations are not recommended. *Severe:* Suggests a combination of ≥2 active agents (polymyxin, aminoglycoside, tigecycline, and sulbactam).	IDSA strongly favours sulbactam-based regimens (new agent preferred) [23]. ESCMID recommendations are more varied based on susceptibility and site, cautioning against some combinations. Neither provides specific duration guidance [37].

Abbreviations: TMP-SMX, trimethoprim-sulfamethoxazole; S, susceptible; R, resistant. Note: This table summarises primary approaches based on cited snippets; consult full guidelines for complete details and nuances.

**Table 2 ijms-26-06905-t002:** Summary of key evidence comparing short versus long antibiotic duration by infection site.

Infection Site	Key Study/Evidence Source (Snippets)	Typical Durations Compared	Main Finding (Short vs. Long)	Key Outcomes Assessed	Major Caveats/Limitations
**BSI**	Yahav et al. [54], Von Dach et al. [55], Molina et al. [56], The BALANCE Investigators et al. [58].	7 d vs. 14 d	Non-inferiority of 7 d	Mortality (30/90 d), relapse, LOS, readmission, and adverse events	Primarily uncomplicated *Enterobacterales* BSI (often urinary source); adequate source control and clinical stability required; MDR pathogens, immunocompromised, andcritically ill often excluded/underrepresented; clinician reluctance [59].
**HAP/** **VAP**	Chastre et al. [75], Pugh et al. [76], Dimopoulos et al. [77,78], IDSA/ATS, ERS/ESCMID Guidelines [79,80,81,82].	7–8 d vs. 10–15 d	Non-inferiority of 7–8 d (overall)	Mortality, clinical cure, recurrence/relapse, and antibiotic-free days	Higher recurrence/relapse risk noted with short course for NF-GNB VAP [61]; optimal duration for MDR NF-GNB VAP remains debated [64].
**cUTIs/** **Pyelonephritis**	van Nieuwkoop et al. [83], Drekonja et al. [66], Germanos et al. [84].	5–7 d vs. 10–14 d	Non-inferiority of 7 d often found for *Enterobacterales* (including ESBL-E) and clinically stable patients	Symptom resolution, mortality, and recurrence/reinfection	Longer duration often used for *P. aeruginosa* [64]; 7 d efficacy in cUTI + bacteraemia may depend on IV/high bioavailability agents [68]; prostatitis requires longer therapy [64]; limited data for CRE/DTR-*P. aeruginosa*.
**IAIs**	Sawyer et al. [71], Sartelli et al. [74].	Fixed 4 d vs. 8 d (until clinical resolution)	Non-inferiority of fixed 4d	Composite (SSI, recurrent IAI, and death)	Requires adequate source control [85]; primarily community-acquired IAI studied [72]; limited data for MDR-GNB IAI or incomplete source control [73].

Abbreviations: LOS, length of stay; SSI, surgical site infection. Note: this table provides a high-level summary; specific study designs, populations, and detailed results vary.

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
