# Peer review of "Antibiotic Therapy Duration for Multidrug-Resistant Gram-Negative Bacterial Infections: An Evidence-Based Review"

_ijms, 2025, doi:10.3390/ijms26146905_

Round 1
Reviewer 1 Report
Comments and Suggestions for Authors
The authors reviewed current evidence and guidelines regarding antibiotic duration for MDR-GNB infections, and compared shorter versus longer antibiotic courses in clinical. There are a few questions I’d like the authors to answer:
- Does Figure 1 have citations? If do, please add citations in the legend.
- In section 4. The MDR-GNB Landscapes, subtitles can be added to improve the layout, eg: 4.1 Extended-spectrum β-lactamase-producing Enterobacterales (ESBL-E).
- In section 6. Evidence Review, there’s no need for subtitle 6.1.
- In section 9. Knowledge Gaps and Future Directions, the authors can add a table to summarize specific gaps and directions to make it more organized.
Author Response
- Does Figure 1 have citations? If do, please add citations in the legend.
Reply: We added an explicative legend
- In section 4. The MDR-GNB Landscapes, subtitles can be added to improve the layout, eg: 4.1 Extended-spectrum β-lactamase-producing Enterobacterales (ESBL-E).
Reply: We prefer not to do that to avoid confusion in the text
- In section 6. Evidence Review, there’s no need for subtitle 6.1.
Reply: We removed it as you suggested
- In section 9. Knowledge Gaps and Future Directions, the authors can add a table to summarize specific gaps and directions to make it more organized.
Reply: We think that it could weigh down the text (too many tables).
Reviewer 2 Report
Comments and Suggestions for Authors
This narrative review addresses a critically important and clinically urgent topic: optimizing antibiotic duration for multi-drug-resistant Gram-negative bacteria (MDR-GNB) infections. The abstract effectively highlights the core challenge—balancing therapeutic efficacy against toxicity and antimicrobial resistance (AMR) escalation—in an area where evidence is notably scarce.
The scope is well-defined and clinically relevant, focusing on high-stakes infections (BSI, HAP/VAP, cUTI, IAI) and key pathogens (CRE, DTR-Pa, CRAB). The authors correctly identify a significant guideline gap: while major societies (IDSA/ESCMID) provide detailed agent selection advice, they lack concrete recommendations on duration. This underscores the review’s value in synthesizing existing evidence to guide practice.
The focus on antimicrobial stewardship goals is timely and necessary. By framing duration optimization as a tool to combat AMR, the review aligns with global health priorities. The abstract is concise, logically structured, and academically rigorous.
Minor Enhancement Suggestion: the full manuscript could further strengthen impact by briefly discussing emerging tools (e.g., biomarkers, PK/PD monitoring) that may aid duration individualization.
Overall, this review fills a vital evidence void and provides a nuanced framework for clinicians navigating therapeutic uncertainty. Its emphasis on stewardship, individualized care, and urgent research needs makes it highly valuable to infectious disease specialists, intensivists, and antimicrobial stewardship teams. I recommend acceptance for publication after standard editorial review.
Author Response
This narrative review addresses a critically important and clinically urgent topic: optimizing antibiotic duration for multi-drug-resistant Gram-negative bacteria (MDR-GNB) infections. The abstract effectively highlights the core challenge—balancing therapeutic efficacy against toxicity and antimicrobial resistance (AMR) escalation—in an area where evidence is notably scarce.
The scope is well-defined and clinically relevant, focusing on high-stakes infections (BSI, HAP/VAP, cUTI, IAI) and key pathogens (CRE, DTR-Pa, CRAB). The authors correctly identify a significant guideline gap: while major societies (IDSA/ESCMID) provide detailed agent selection advice, they lack concrete recommendations on duration. This underscores the review’s value in synthesizing existing evidence to guide practice.
The focus on antimicrobial stewardship goals is timely and necessary. By framing duration optimization as a tool to combat AMR, the review aligns with global health priorities. The abstract is concise, logically structured, and academically rigorous.
Minor Enhancement Suggestion: the full manuscript could further strengthen impact by briefly discussing emerging tools (e.g., biomarkers, PK/PD monitoring) that may aid duration individualization.
Overall, this review fills a vital evidence void and provides a nuanced framework for clinicians navigating therapeutic uncertainty. Its emphasis on stewardship, individualized care, and urgent research needs makes it highly valuable to infectious disease specialists, intensivists, and antimicrobial stewardship teams. I recommend acceptance for publication after standard editorial review.
Reply: Thank you for your valuable comments. We add a sub-paragraph 8.1 discussing what you suggested.